



**Distinct evolutions of haze pollution from winter to following spring over the**
**North China Plain: Role of the North Atlantic sea surface temperature anomalies**
Linye Song[1], Shangfeng Chen[2*], Wen Chen[2], Jianping Guo[3], Conglan Cheng[1], and
Yong Wang[4]
[1]*Institute of Urban Meteorology, China Meteorological Administration, Beijing,*
*China*
[2]*Center for Monsoon System Research, Institute of Atmospheric Physics, Chinese*
*Academy of Sciences, Beijing, China*
[3]*State Key Laboratory of Severe Weather, Chinese Academy of Meteorological*
*Sciences, Beijing, China*
[4]*ZAMG, Central Institute for Meteorology and Geodynamics, Vienna, Austria*
*Atmospheric Chemistry and Physics*
Submitted in March 2021
Corresponding authors:
Shangfeng Chen
E-mail: chenshangfeng@mail.iap.ac.cn



**Abstract**

This study reveals that haze pollution (HP) over the North China Plain (NCP) in
winter can persist to following spring during most years. The persistence of HP$_{NCP}$ is
attributed to maintenance of an anticyclonic anomaly (AA) over northeast Asia and
southerly wind anomalies over the NCP. Southerly wind anomalies over the NCP
reduce surface wind speed and increase relative humidity, which are conducive to
above-normal HP$_{NCP}$ both in winter and spring. However, there exist several years
when above-normal HP$_{NCP}$ in winter are followed by below-normal HP$_{NCP}$ in the
following spring. The reversed HP$_{NCP}$ in winter and spring in these years is due to the
inverted atmospheric anomalies over northeast Asia. In particular, AA over northeast
Asia in winter is replaced by a cyclonic anomaly (CA) in the following spring. The
resultant spring northerly wind anomalies over NCP are conducive to below-normal
HP$_{NCP}$. These two distinctive evolutions of HP$_{NCP}$ and atmospheric anomalies over
northeast Asia from winter to spring are attributed to the different evolutions of sea
surface temperature anomalies (SSTA) in the North Atlantic. In the persistent years,
warm North Atlantic SSTA in winter maintains to following spring via positive air-sea
interaction process and induces a negative spring North Atlantic Oscillation
(NAO)-like pattern, which contributes to the AA over northeast Asia via atmospheric
wave train. By contrast, in the reverse years, cold SSTA in the North Atlantic is
maintained from winter to spring, which induces a positive spring NAO-like pattern
and leads to CA over northeast Asia via atmospheric wave train. The findings suggest
that North Atlantic SSTA plays crucial roles in modulating the distinct evolutions of



HP$_{NCP}$ from winter to succedent spring, which can be served as an important
preceding signal for haze pollution evolution over the North China Plain.
**Keywords**: Evolution of Haze pollution; North China Plain; North Atlantic sea
surface temperature; North Atlantic Oscillation; Atmospheric circulation



## 1. Introduction

Haze pollution has become a serious air quality issue in China accompanying the rapid urbanization and fast economic development (e.g. Ding and Liu 2004; Wang and Chen 2016; Zhang et al. 2018). It has been well recognized that the occurrences of haze pollution event can exert substantial impacts on the human health, air transportation, ground traffic, agriculture production, and regional climate change (e.g. Koren et al. 2012; Zhang and Crooks 2012; Fu et al. 2014; Wang et al. 2014a, 2014b; Wu et al. 2016; Tie et al. 2016; Cohen et al. 2017; Guo et al. 2018; Zhang et al. 2018; Lu et al. 2019). For example, Cohen et al. (2017) reported that near 4.2 million premature deaths in the world in 2015 were attributed to the overexposure of PM2.5. In addition, haze pollution is suggested to result in a decrease of about 1.2%-3.8% of the annual Gross National Product (GNP, Zhang and Crooks 2012). Furthermore, increasing concentration of anthropogenic aerosol, which is related to the enhanced haze pollution, could exert significant impacts on the atmospheric circulation and regional precipitation change (Koren et al. 2012; Wang et al. 2014). Considering the notable impacts of haze pollution, it is of great scientific importance to improve our understanding of the factors contributing to haze pollution and the associated mechanisms.

A number of previous studies have investigated the factors responsible for the variations of haze pollution in China on multiple timescales. The long-term increasing trend of haze pollution in China is generally attributable to the rapid increases in anthropogenic emissions (e.g. Che et al. 2009; Ding and Liu 2014; Zhao et al. 2016;



Cheng et al. 2019). For example, Zhao et al. (2016) showed that the notable
increasing trend of haze pollution in winter over eastern China has a close relationship
with the Gross Domestic Product (GDP) in China. Several studies suggested that
changes in the meteorological conditions due to global warming also play a role in the
long-term trend of haze pollution in China (e.g. Cai et al. 2017; Liu et al. 2017; Ding
et al. 2017; Zhang et al. 2020).

On the interannual and interdecadal timescales, variations of the haze pollution

in China are suggested to be mainly controlled by the meteorological conditions. For
instance, Dang and Liao (2019) reported that the changes of meteorological
conditions accounted for about 70% of the variation of the annual haze days in the
Beijing-Tianjin-Hebei region. Zhao et al. (2016) suggested that the Pacific Decadal
Oscillation could exert marked impacts on the interdecadal variation of the haze
pollution in eastern China via inducing large-scale atmospheric circulation anomalies
over East Asia. Pacific Decadal Oscillation is the first leading mode of sea surface
temperature anomalies (SSTA) in the North Pacific on the interdecadal timescale
(Mantua et al., 1997; Zhang et al., 1997; Duan et al. 2013). Xiao et al. (2014) showed
that the Atlantic Multidecadal Oscillation modulates haze pollution in China via
triggering atmospheric wave train over Eurasia. Atlantic Multidecadal Oscillation is
the dominant mode of SSTA in the North Atlantic on the multidecadal timescale (Kerr
2000). Compared to the interdecadal variation, much more studies have examined the
factors for the interannual variation of haze pollution in China, mainly concentrating
on boreal winter. It is shown that interannual variation of haze pollution in eastern



China can be impacted by the Arctic Oscillation (Yin et al. 2015), East Asian winter
Monsoon (Li et al. 2016; Chen et al. 2020), El Niño-Southern Oscillation (Guo and Li
2015; Chang et al. 2016; Liu et al. 2017; Li et al. 2017; He et al. 2019), North Atlantic
SSTA (Xiao et al. 2014), Arctic sea ice (Wang et al. 2015; Yin and Wang 2017),
Eurasian snow cover (Yin and Wang 2018), and the East Atlantic-Western Russian
(EAWR) teleconnection pattern (Yin and Wang 2017; Chen et al. 2020). A recent
study has examined the factors modulating the interannual variation of springtime
haze pollution in the North China Plain Region (NCPR) (Chen et al. 2019). Note that
NCPR is one of the most important regions in China with very dense population, large
traffic activities and highly developed economy. In addition, NCPR is also the most
polluted region in China (Yin et al. 2015). Chen et al. (2019) indicated that North
Atlantic SSTA and the North Atlantic Oscillation (NAO, the first leading mode of
interannual atmospheric variability over the North Atlantic region; Hurrell 1995) play
important roles in determining the haze pollution over NCPR via modulating
atmospheric circulation anomalies over northeast Asia through triggering atmospheric
wave train extending from North Atlantic across Europe to East Asia (Chen et al.
2019). Previous studies mainly investigated interannual variations of haze pollution
over the NCPR in winter and spring separately. However, impacts of haze pollution
may depend strongly on the time period of persistence. Hence, an important question
is raised: whether there exists a relation between interannual variation of haze
pollution over the NCPR in winter and following spring? In particular, could the
wintertime haze pollution maintain from winter to the following spring? If so, what



are the plausible factors contributing to the across-season persistence of haze
pollution over the NCPR from winter to succedent spring? Understanding the
evolution features of the haze pollution from winter to spring and the associated
mechanisms would have important implications for the seasonal prediction of haze
pollution over the NCPR. In this study, the issues raised above will be investigated
and addressed.
The remainder of this paper is organized as follows. Section 2 describes the data
and methods used in this study. Section 3 examines relation of interannual variations
between winter and spring haze pollution over the NCPR, and compares the two
distinct types of haze evolutions found in this paper. Section 4 examines the factors
responsible for the different evolutions of haze pollution over NCPR from winter to
the following spring. Summary and discussion are provided in section 5.

**2. Data and methods**
*2.1 Data*
Monthly mean horizontal winds, geopotential height, relative humidity, surface
wind speed, surface heat fluxes are obtained from the National Centers for
Environmental Prediction-National Center for Atmospheric Research (NCEP-NCAR)
reanalysis                (Kalnay                et al.                1996;
https://psl.noaa.gov/data/gridded/data.ncep.reanalysis.html), which are available from
January 1948 to the present. Surface heat fluxes are the sum of the surface latent and
sensible heat fluxes, surface shortwave and longwave radiations. Atmospheric data



from the NCEP-NCAR reanalysis have a horizontal resolution of 2.5°×2.5° in the
longitude-latitude grids, while surface heat fluxes are on T62 Gaussian grids. Monthly
mean SST data are derived from the National Oceanic and Atmospheric
Administration (NOAA) Extended Reconstructed SST version 5 (ERSSTV5) from
January    1854    to    the    present    (Huang    et    al.    2017;
https://psl.noaa.gov/data/gridded/data.noaa.ersst.v5.html), with a horizontal resolution
of 2°×2° in the longitude-latitude grids. Atmospheric teleconnection indices,
including the EAWR index and NAO index, are provided by the NOAA Climate
Prediction Center (https://www.cpc.ncep.noaa.gov/data/).

Surface data of visibility and relative humidity observed at 748 meteorological

stations are extracted from the National Meteorological Information Center of China
from 1979 to 2012. These meteorological observations are measured daily at 0200,
0800, 1400 and 2000 Beijing time (UTC+8). Following previous studies (Guo et al.
2017; Chen et al. 2019, 2020), a series of quality control techniques are applied to this
meteorological data to ensure its quality and consistency. In particular, we exclude the
observation station if it contains any missing values throughout the whole analysis
period. In addition, the data has been removed when precipitation, snow events, and
dust storms occurred. After the above quality control, there remain 218 stations over
Eastern China (Fig. 1a). Furthermore, following previous studies (Che et al. 2009;
Guo et al. 2017, Chen et al. 2019; 2020), we only use the data at 1400 Beijing time, as
this time may be the most representative of the daily visibility compared to other
times. It should be mentioned that the atmospheric visibility, which is traditionally



measured by human visual observation, starts to be determined by the automatic
visibility instruments since the year 2014. Due to the changes of the observation
methods, large uncertainties have emerged due to the issues of heterogeneity as
reported by Li et al. (2018). Thus, this study does not employ the visibility data after
the 2014.
Long-term trends of all variables have been removed to avoid the impact of the
global warming signal and to focus on the interannual variation of haze pollution.
Anomalies are calculated by subtracting the monthly climatology from the original
data. Significance levels of correlation coefficient and composite differences are
estimated based on the two-tailed Student's $t$ test.

***2.2 Dry Extinction coefficient of aerosol***
As in previous studies (Li et al. 2018; Guo et al. 2017; Chen et al. 2019, 2020),
this study employs the dry extinction coefficient (DECC) of aerosol to represent the
haze pollution. The DECC is defined based on the Koschmieder relationship
(Koschmieder 1926):
$$DECC = \frac{K}{Vis_{dry}} \qquad (1)$$

where $K$ is equal to 3.912, $Vis_{dry}$ indicates atmospheric visibility after removing the
effect of relative humidity. It is noted that atmospheric visibility is not only impacted
by the dry particles, but also affected by the amount of water uptake. For instance,
high humidity associated with fog could lead to reduction of atmospheric visibility.
Previous studies suggested that the visibility needs to be corrected in the presence of a



relative humidity ranging from 40 to 90% (e.g. Rosenfeld et al. 2007), which is
expressed as follows:
$$Vis_{dry} = \frac{Vis_{obs}}{0.26 + 0.4285\log(100 - RH)} \qquad (2)$$

where $Vis_{obs}$ indicates the observed visibility. Note that all visibility data are discarded
when the relative humidity (RH) is higher than 90% to remove the influence of fog
events, non-linear aerosol and water interactions (Craig and Faulkenberry 1979; Guo
et al. 2017; Chen et al. 2019, 2020).

*2.3 Wave activity flux*
We use the wave activity flux defined by Takaya and Nakamura (2001) to
examine the stationary Rossby wave propagation, which can be expressed as follows:
$$W = \frac{1}{2|U|}\begin{pmatrix} U\left(v'^2 - \psi'v'_x\right) + V\left(-u'v' + \psi'u'_x\right) \\ U\left(-u'v' + \psi'u'_x\right) + V\left(u'^2 + \psi'u'_y\right) \\ \frac{f_oR_ap}{N^2H_o}\left\{U\left(v'T' - \psi'T'_x\right) + V\left(-u'T' - \psi'T'_y\right)\right\} \end{pmatrix} \qquad (3)$$

where $\boldsymbol{U} = (U, V)$ is the climatological wind vector. $\boldsymbol{V} = \left(u', v'\right)$ denotes
geostrophic winds anomalies. $\psi'$ is geostrophic stream function anomalies. $H_o$, $p$,
and $N$ represent scale height, pressure normalized by 1000-hPa, and Brunt-Vaisala
frequency, respectively. $R_a$, $T'$, and $f_o$ denote gas constant of the dry air, air
temperature anomalies, and the Coriolis parameter at 45°N, respectively. Subscripts x
and y are the derivatives in the zonal and meridional directions, respectively.
Climatological mean is calculated over the 1980–2010 period.



*2.4 Barotropic model*
The present study employs a linear barotropic model to investigate the role of the
SST anomalies (SSTA) over the subtropical and tropical North Atlantic in triggering
atmospheric Rossby wave train over mid-high latitudes of Eurasia. Previous studies
have demonstrated that cold (warm) SSTA in the subtropical and tropical regions are
able to induce convergence (divergence) anomalies at the upper-troposphere that act
as effective sources of atmospheric stationary Rossby wave (Hodson et al. 2010;
Watanabe 2004; Zuo et al. 2013; Wu et al. 2011; Chen et al. 2016, 2020). Based on a
simple barotropic vorticity equation (Watanabe 2004; Sardeshmukh and Hoskins 1988;
Chen et al. 2020), the barotropic model is established by:
$$\partial_t \nabla^2 \psi' + J\left(\overline{\psi}, \nabla^2 \psi'\right) + J\left(\psi', \nabla^2 \overline{\psi} + f\right) + \alpha \nabla^2 \psi' + \nu \nabla^6 \psi' = S' \qquad (4)$$
where $\psi'$ and $\overline{\psi}$ are the perturbation stream function and basic state stream
function, respectively. $f$ and $J$ represent the Coriolis parameter and Jacobian
operator, respectively. $S'$ represents the vorticity source generated by the atmospheric
divergence. The barotropic model consists a biharmonic diffusion and a linear
damping that indicate the Rayleigh friction. Note that solution of the above Equation
associated with the barotropic model is determined by the vorticity perturbation ($S'$)
and the basic state. In the present analysis, the basic state is chosen at the 300-hPa
level over 1979-2010 using the NCEP-NCAR reanalysis data. O'Reilly et al. (2018)
reported that results of the barotropic model experiments are insensitive to the basic
states chosen from the upper troposphere (e.g., from 350-hPa to 200-hPa). It should
be mentioned that the basic state is chosen from the upper troposphere because the





strongest convergence/divergence anomalies generated by the tropical and subtropical
SST cooling/warming tend to be observed at the upper troposphere (e.g., Sun et al.
2015; Krishnamurti et al. 2013; O'Reilly et al. 2018; Chen et al. 2020).

**3. Connection of haze pollution over NCPR in winter and spring**
Following previous analyses (Yin and Wang 2016; Chen et al. 2019, 2020), the
NCPR corresponds to the region spanning 34°–43°N, 114°–120°E. Slight changes of
the region to represent NCPR don't affect the main results of this study. Figure 1a
shows that there are a total of 26 meteorological observational stations in the NCPR
(red dots in the box). As in previous studies (Yin and Wang 2016; Chen et al. 2019,
2020), this analysis defines a NCPR DECC index (NDI for short) by averaging the
DECC anomalies over the above 26 stations to describe variation of haze pollution
over the NCPR.
Figure 1b shows year-to-year variations of the NDI in winter and the following
spring over 1980–2011. The correlation coefficient between the winter and spring
NDI over 1980–2011 is 0.30, exceeding the 90% confidence level, which suggests a
marginal in-phase variation of the haze pollution in winter and the following spring.
In particular, most of the positive (negative) values of winter NDI are followed by
extremely large (small) values of the spring NDI (for example, years in 1980, 1985,
1986, etc.). This suggests that air condition over the NCPR in the following spring
tends to be better (worse) than normal if haze pollution in preceding winter is less
(more) serious over the NCPR. As shown in Fig. 1b, however, there also exists several
years when values of the winter and following spring NDI are strongly opposite,
showing out-of-phase variation. In these years, large negative (positive) values of
winter NDI are followed by large positive (negative) spring NDI (Fig. 1b). For
instance, in 1984 and 1991, the winter NDIs are significantly negative, while the
following spring NDIs are significantly positive. In 1982 and 1989, large positive
values of winter NDI are followed by large negative values of spring NDI.

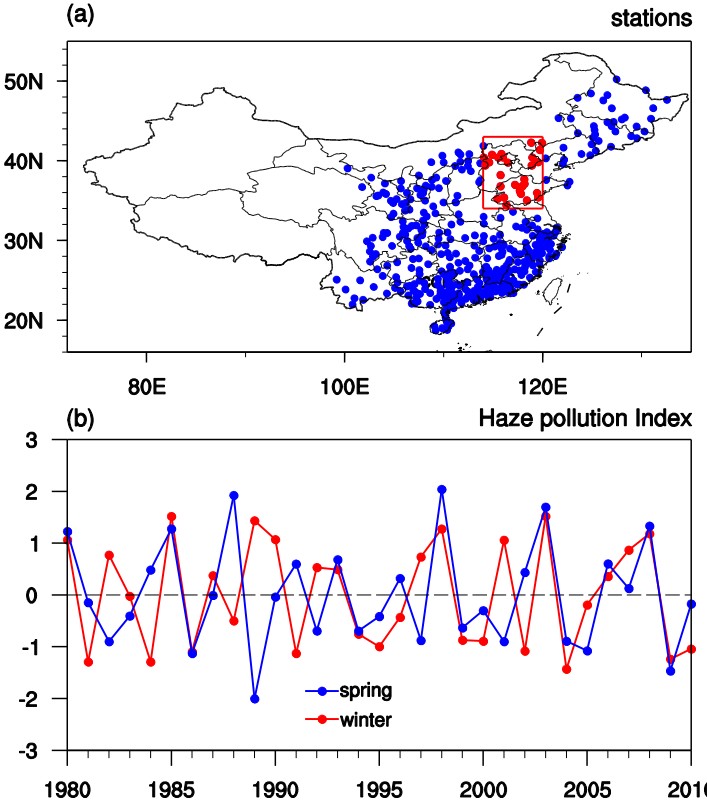

**Figure 1**. (a) Geographical locations of the meteorological stations (denoted by dots)
in China. Red dots represent the meteorological stations in the NCPR. (b)
Standardized time series of the NDI in winter (December-January-February-mean,
D(0)JF(1) for short) and its following spring (March-April-May-mean, MAM(1) for
short) over 1980-2010.



**Table 1**. Lists of the persistent and reverse years.

| Persistence (11 years) | Reverse (9 years) |
|---|---|
| 1980, 1985, 1986, 1993, 1994, 1998, 1999, 2003, 2004, 2008, 2009 | 1982, 1984, 1988, 1989, 1991, 1992, 1997, 2001, 2002 |

In the following, positive (negative) phases of the winter and spring NDIs are
identified when the normalized NDIs are larger (less) than 0.43. Previous studies
indicated that value of $\pm 0.43$ standard deviation can separate a time series into three
portions (positive and negative phases, and normal condition) with nearly the same
sample sizes. Note that a use of $\pm 0.5$ standard deviation as the threshold to define
anomalous NDI years leads to similar results, but with smaller sample sizes. Table 1
presents the years when winter and spring NDIs are in-phase and out-of-phase.
According to Table 1, there are a total of 11 (9) years for the in-phase (out-of-phase).
Relatively less number of out-of-phase years than in-phase years is found during
1980–2011, in concert with the evidence that winter NDI has a marginal positive
correlation with the spring NDI. In the following, out-of-phase (in-phase) years are
called (reverse) persistent years for convenience of descriptions. We employ
composite analysis to compare evolutions of DECC and atmospheric anomalies
between the persistent and reverse years. Note that in performing the composite
analysis, we reversed the anomalies when the winter NDI is negative since, to a large
extent, the DECC and atmospheric circulation anomalies over the NCPR are
symmetric between the positive and negative phases of the winter NDI. Hence, the
descriptions below are corresponding to the positive phases of the winter NDI, but
also apply to the negative phases except with opposite signs.

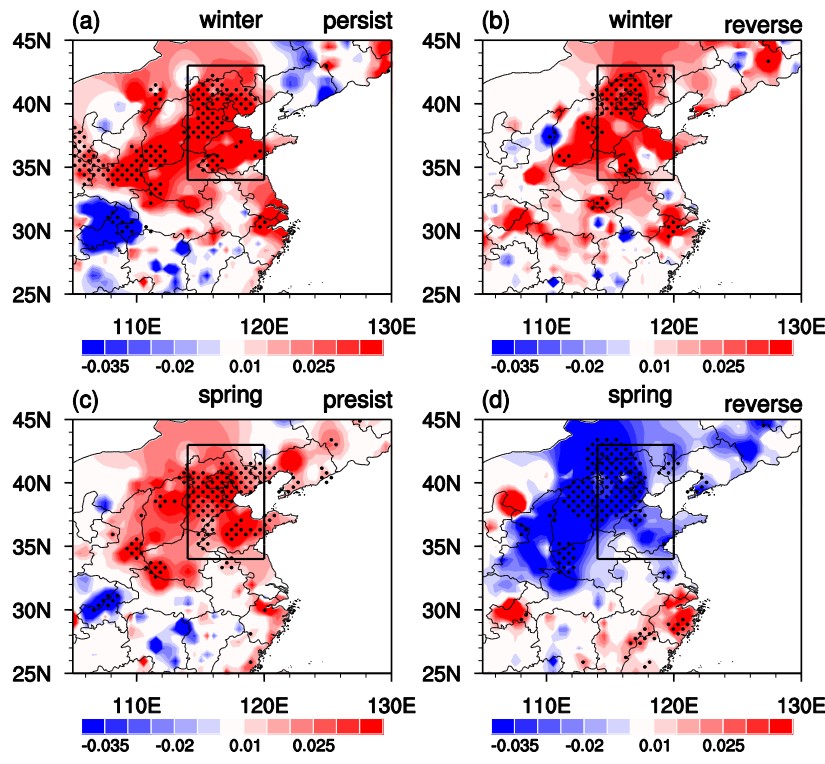

**Figure 2**. Composite anomalies of DECC (unit: km$^{-1}$) in (a, b) winter and (c, d) spring
in the (left column) persistent years and the (right column) reverse years. Stippling
regions indicate anomalies significant at the 5% level.

Figure 2 shows composite anomalies of DECC in winter and following spring in

the persistent and reverse years. For the persistent years, large positive DECC
anomalies (indicating more serious haze pollution) are seen over the NCPR and
surrounding regions (Fig. 2a). DECC anomalies in winter over southern China are
weak, suggesting a weak relation of the haze pollution between northern and southern
China, consistent with previous studies (e.g. Li et al. 2017; He et al. 2019). Large
positive DECC anomalies over the NCPR are maintained to the succedent spring with
comparable amplitude (Figs. 2a and 2c). For the reverse years, large positive DECC
anomalies also appear over the NCPR in winter (Fig. 2b). However, in the following
spring, the NCPR and surrounding regions are dominated by significantly negative
values of DECC (Fig. 2d) (indicating air condition in spring becomes better), which is
in sharp contrast to that for the persistent years (Fig. 2c).

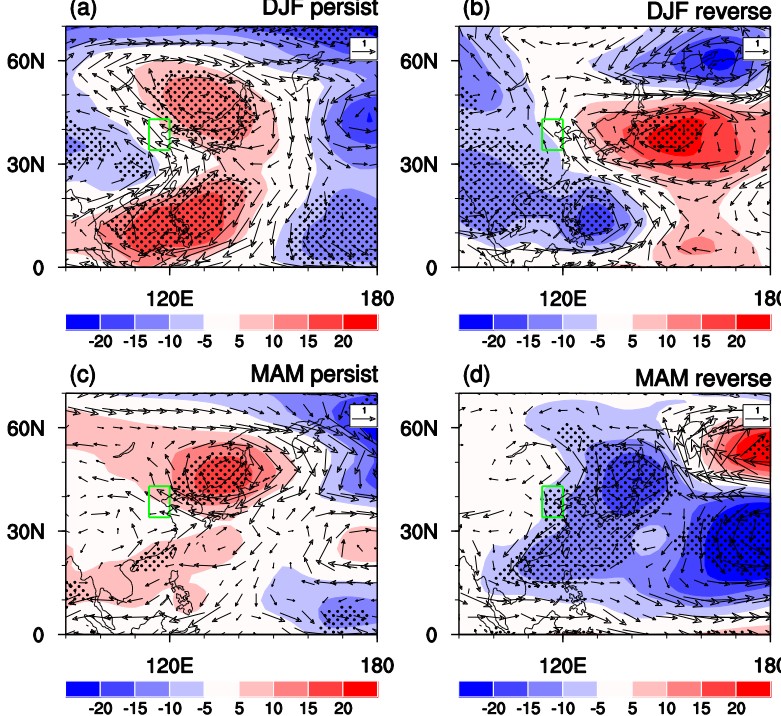

**Figure 3**. Composite anomalies of 850-hPa winds (vectors, unit: m s$^{-1}$) and
streamfunction (shadings; unit: $10^5$ m$^2$ s$^{-1}$) in (a, b) winter and (c, d) spring in the (left
column) persistent years and the (right column) reverse years. Stippling regions
indicate streamfunction anomalies that are statistically significant at the 5% level.

Studies have demonstrated that meteorological conditions related to the

atmospheric anomalies can explain above 66% of interannual and interdecadal
variations of haze pollution over most parts of Eastern China (Zhang et al. 2014; Chen



et al. 2019; He et al. 2019; Dang and Liao 2019; Ma and Zhang 2020). Hence, it is
expected that different evolutions of the DECC anomalies from winter to following
spring over the NCPR may be associated with the distinct evolutions of atmospheric
anomalies. Composite anomalies of winds and streamfunction at 850hPa in winter and
following spring for the persistent and reverse years are shown in Fig. 3. In the
persistent years, a significant anticyclonic anomaly is seen over northeast Asia,
accompanied by strong southerly winds anomalies over NCPR, and northerly wind
anomalies over mid-latitudes North Pacific (Fig. 3a). In addition, another marked
anticyclonic anomaly appears over south China sea and Philippine sea, leading to
strong southerly wind anomalies over southern China (Fig. 3a). The anomalous
anticyclone over northeast Asia and associated southerly wind anomalies over NCPR
are maintained to following spring (Fig. 3c).
For the reverse years, a strong anticyclonic anomaly also exists over northeast
Asia, but with a southeastward displacement (Fig. 3b) compared to that in the
persistent years. Note that NCPR is also dominated by strong southerly wind
anomalies (Fig. 3b). In contrast, the south China sea and Philippine sea are covered by
an anomalous cyclone, together with northerly wind anomalies over southern China
(Fig. 3b). Moreover, an anticyclonic anomaly occurs around the Russian Far East,
accompanied by westerly wind anomalies to the north of Japan (Fig. 3b). In the
following spring, the pronounced anticyclonic anomaly over northeast Asia and
associated southerly wind anomalies over the NCPR are replaced by a marked
cyclonic anomaly and northerly wind anomalies (Fig. 3d).
Hence, there appears prominent difference in the atmospheric anomalies over
northeast Asia between the persistent and reverse years. Specifically, in the persistent
years, the anomalous anticyclone over northeast Asia and southerly wind anomalies
over NCPR are maintained from winter to following spring. By contrast, in the
reverse years, the wintertime anticyclonic anomaly is replaced by a cyclonic anomaly
over northeast Asia, accompanied by reversal of meridional wind anomalies over
NCPR from winter to the succedent spring.

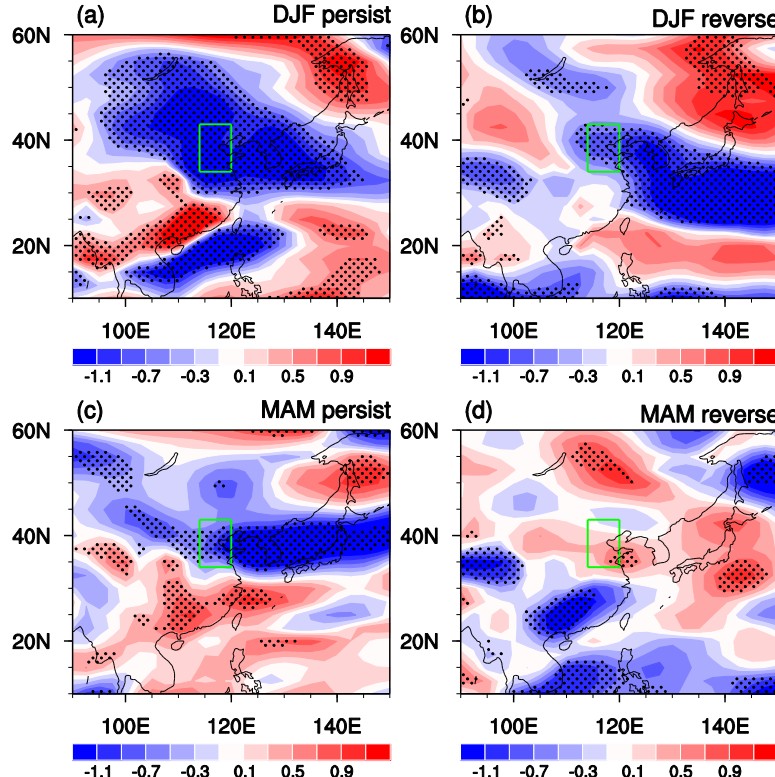


**Figure 4**. Composite anomalies of 850-hPa wind speed (unit: m s$^{-1}$) in (a, b) winter
and (c, d) spring in the (left column) persistent years and the (right column) reverse
years. Stippling regions indicate anomalies that are statistically significant at the 5%
level.

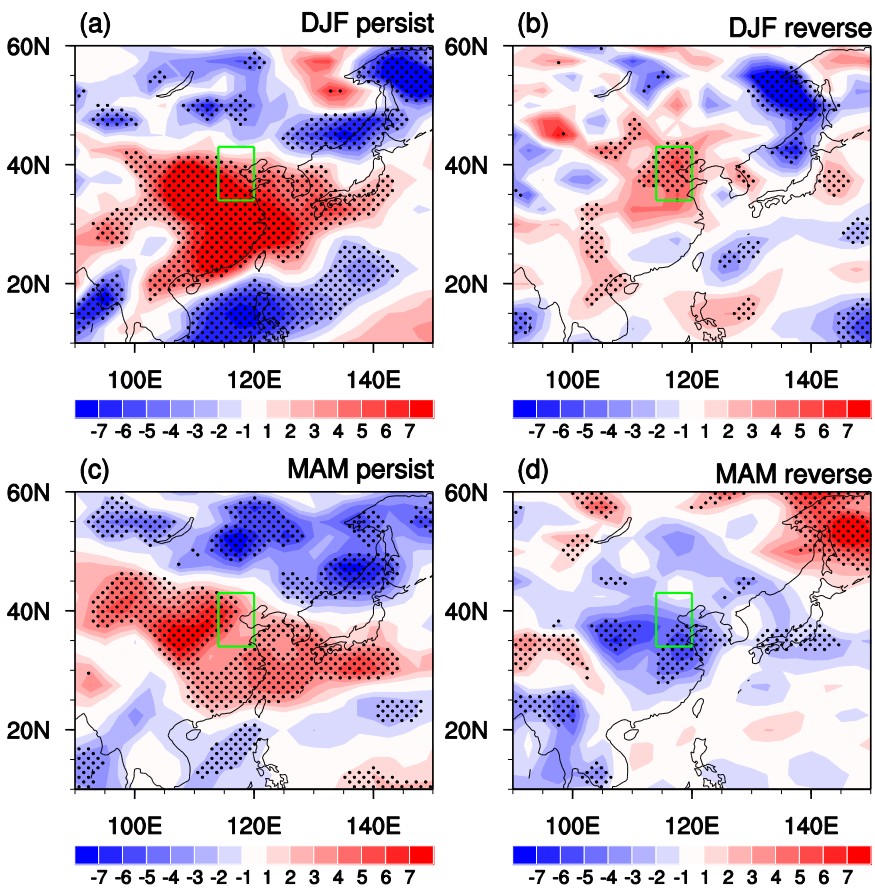

**Figure 5**. Composite anomalies of 850-hPa relative humidity (unit: %) in (a, b) winter and (c, d) spring in the (left column) persistent years and the (right column) reverse years. Stippling regions indicate anomalies that are statistically significant at the 5% level.

The atmospheric anomalies can impact haze pollution over NCPR in winter and spring via modulating surface wind speed and relative humidity (e.g. Hang et al. 2013; Chen et al. 2019, 2020; He et al. 2019; Dang and Liao 2019; Li et al. 2020). The increase (decrease) in the surface wind speed is (not) conducive to horizontal diffusion of pollutants, thus contributing to below (above) normal DECC and less (more) serious haze pollution (Chen et al. 2019; Li et al. 2020). Additionally, large



(small) relative humidity is (not) conducive to the generation of secondary organic
compounds and secondary aerosol species (such as $SO_4^{2-}$ and $NO_3^-$), which contribute
to more (less) serious haze pollution over NCPR (Yu et al. 2005; Hennigan et al. 2008;
Chen et al. 2019; Li et al. 2020; Ma and Zhang 2020).

Composite anomalies of low-level (850-hPa) wind speed and relative humidity in

winter and spring are shown in Fig. 4 and Fig. 5, respectively. In winter, low-level
wind speed is significantly decreased over the NCPR with a northwestward extension
to the Lake Baikal and an eastward extension to western North Pacific for both the
persistent and reverse years (Figs. 4a and 4b). The southerly wind anomalies to the
western side of the anticyclonic anomaly over northeast Asia (Figs. 3a and 3b) as
opposite to the climatological northerly winds dominated by East Asian winter
monsoon (not shown), lead to decreases in the total wind speed (Figs. 4a and 4b),
which contributes to more serious haze pollution (Figs. 2a and 2b). In addition,
southerly winds anomalies tend to bring more water vapor northward from Southern
Ocean and result in an increase in the relative humidity (Fig. 5a and 5b), which are
also conducive to formation of secondary aerosol species (Yu et al. 2005; Hennigan et
al. 2008; Chen et al. 2019) and contribute to more serious haze pollution over NCPR
in winter (Figs. 2a and 2b). In the persistent years, sustenance of the anticyclonic
anomaly over northeast Asia and southerly wind anomalies over NCPR to the
following spring (Fig. 3c) contributes to above normal DECC in spring (Fig. 2c) via
reducing surface wind speed (Fig. 4c) and increasing relative humidity (Fig. 5c). By
contrast, in the reverse years, reversal of atmospheric anomalies over northeast Asia





from anticyclonic anomaly in winter (Fig. 3b) to cyclonic anomaly in the following
spring (Fig. 3d) results in the inverted DECC anomalies over NCPR (Figs. 2b and 2d).
In spring, the northerly wind anomalies increase the low-level total wind speed (Fig.
4d), which are conducive to the horizontal dispersion of pollutants and contribute to a
better air condition. Additionally, the anomalous northerly winds (Fig. 3d) lead to
decrease in the relative humidity (Fig. 5d) via carrying colder and drier air from
higher latitude, suppressing generation of secondary organic compounds and
secondary aerosol species, and also contributing to mitigation of haze pollution (Fig.
2d). Above evidences suggest that the distinct evolutions of haze pollution over
NCPR between the persistent and reverse years are closely related to the different
evolutions of atmospheric anomalies over northeast Asia.

Atmospheric anomalies over northeast Asia related to interannual variations of

haze pollution over NCPR are closely associated with upstream atmospheric wave
train over North Atlantic and mid-high latitudes Eurasia. Studies have demonstrated
that atmospheric wave trains originated from North Atlantic across Eurasia to East
Asia have a strong contribution to interannual variations of haze pollution and climate
anomalies over North China (Yin and Wang 2016, 2017; Zhao et al. 2019; Chen et al.
2019, 2020). Composite anomalies of geopotential height at 500-hPa over larger areas
in winter and succedent spring for the persistent and reverse years are presented in Fig.
6. To examine the possible sources of the atmospheric wave trains over Eurasia, we
also present the wave activity fluxes in Fig. 6, which describe propagation directions
of the atmospheric Rossby waves. Spatial structures of the geopotential height
anomalies at 850-hPa and 200-hPa (not shown) are highly similar to those at 500-hPa
in Fig. 6, indicating a vertically barotropic structure of the atmospheric anomalies
over mid-high latitudes of North Atlantic and Eurasia.

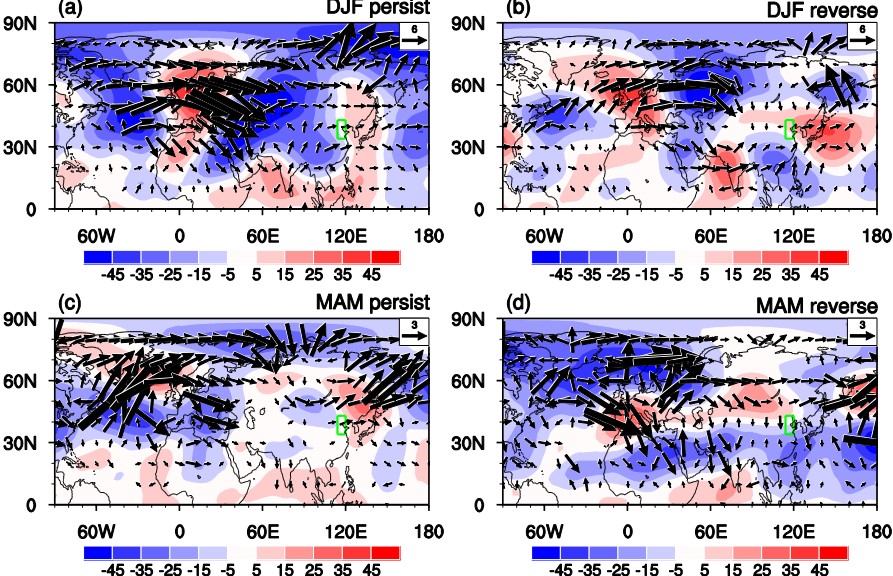


**Figure 6.** Composite anomalies of 500-hPa geopotential height (shadings, unit: m)
and wave activity fluxes (vectors; unit: $m^2\ s^{-2}$) in (a, b) winter and (c, d) spring in the
(left column) persistent years and the (right column) reverse years.
In the persistent years, an EAWR-like teleconnection pattern is obviously
observed extending from North Atlantic across Europe to East Asia, with negative
geopotential height anomalies (corresponding to cyclonic anomalies) over mid-high
latitudes North Atlantic and central Eurasia, and positive geopotential height
anomalies (corresponding to anticyclonic anomalies) over west Europe and northeast
Asia (Fig. 6a). The pattern correlation coefficient between the EAWR-related 500-hPa
geopotential height anomalies and those in Fig. 6a over the North Atlantic and
Eurasian regions (i.e. 20°–90°N and 70°W–130°E) reaches 0.65, significant at the



99.9% confidence level. Hence, in the persistent years, the EAWR teleconnection
contributes largely to the formation of the anticyclonic anomaly over northeast Asia in
winter. In the reverse years, spatial structure of the 500-hPa geopotential height
anomalies over mid-high latitudes of North Atlantic and Eurasia (Fig. 6b) bears a
close resemblance to that for the persistent years (Fig. 6a), also resembles the EAWR
teleconnection pattern. We have also calculated the pattern correlation coefficient
between the 500-hPa geopotential height anomalies in Fig. 6b and those related to the
winter EAWR over the similar region of 20°–90°N and 70°W–130°E. The pattern
correlation coefficient is as high as 0.85, slightly higher than that in the persistent
years (r=0.65), suggesting that the EAWR teleconnection pattern also has a strong
contribution to the formation of the wintertime anticyclonic anomaly over Northeast
Asia and haze pollution over the NCPR in the reverse years. Above results are
consistent with Yin and Wang (2017) and Chen et al. (2020). Yin and Wang (2017)
demonstrated that the EAWR teleconnection is the most important atmospheric wave
train modulating haze pollution over North China. Chen et al. (2020) reported that the
winter EAWR teleconnection have a stable and strong impact on the interannual
variation of haze pollution over the NCPR via calculating the running correlation
coefficients between the winter EAWR index and NDI. Note that there exist several
differences in the spatial structure of the wintertime EAWR teleconnection between
the persistent and reverse years (Figs. 6a and 6b). In particular, the center of negative
geopotential height anomalies over central Eurasia in the persistent years (Fig. 6a) is
stronger and shifts southward compared to that in the revere years (Fig. 6b). In





addition, negative geopotential height anomalies over western North Atlantic extend
more southwestward for the revere years (Figs. 6a and 6b). Differences in the spatial
structure of the winter EAWR between the persistent and reverse years may be partly
due to differences in the background mean circulation (Chen and Wu 2017; Wang et al.
2019). Detailed investigation of the factors for the changes of the spatial pattern of the
winter EAWR is out of the scope of this study. Furthermore, it is interesting to note
that an atmospheric Rossby wave exists over subtropical region propagating along the
subtropical Jet stream to extend from north Africa across south Asia and then turn
northeastward to northeast Asia in the reverse years (Fig. 6b). This subtropical wave
train also has a contribution to the formation of the anticyclonic anomaly over
Northeast Asia and interannual variation of haze pollution over the NCPR as has been
indicated by Chen et al. (2020).
In spring, a negative NAO-like pattern appears over North Atlantic in the
persistent years, featured by negative geopotential height anomalies around 40°-50°N
and positive anomalies over 60°-70°N in the persistent years (Fig. 6c). The pattern
correlation coefficient between the spring NAO-related 500-hPa geopotential height
anomalies and the composted 500-hPa geopotential height anomalies in Fig. 6c over
North Atlantic region (30°–80°N and 20°W–60°W) is as high as -0.75. This result is
consistent with Chen et al. (2019), which indicated that negative (positive) phase of
the spring NAO contributes to formation of an anomalous anticyclone (cyclone) over
Northeast Asia and leads to more (less) serious haze pollution over NCPR via
eastward propagating wave train. However, in the reverse years, there exists a positive



NAO-like pattern over the North Atlantic (Fig. 6d), which is in sharp contrast to that
in the persistent years (Fig. 6c). In particular, the pattern correlation between the
500-hPa geopotential height anomalies in Fig. 6c and spring NAO-related anomalies
over 30°–80°N and 20°W–60°W reaches 0.6. As indicated by Chen et al. (2019), the
spring positive NAO would contribute to below-normal DECC over the NCPR.

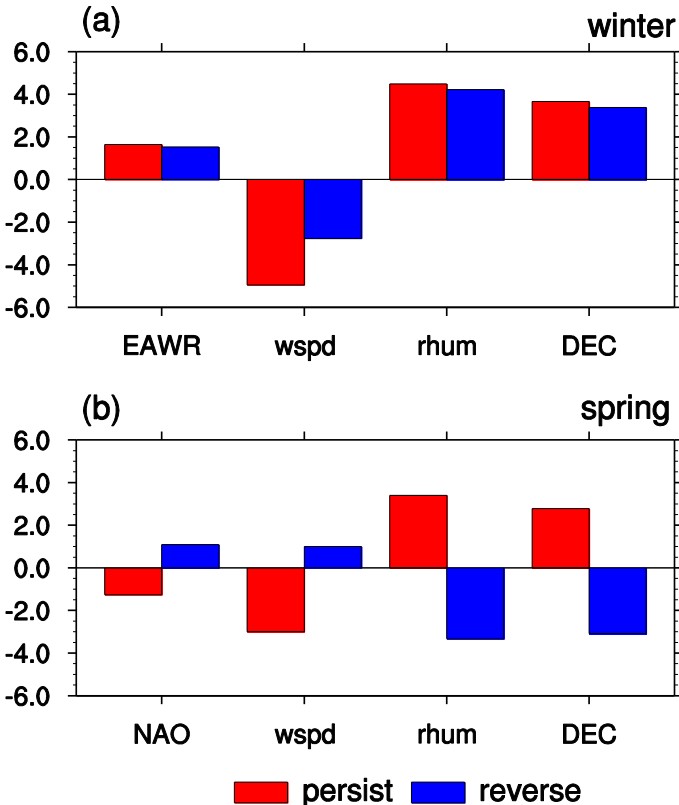


**Figure 7**. (a) Composite anomalies of the EAWR index, wind speed at 850-hPa ($4^{-1}$ m
$s^{-1}$), relative humidity at 850-hPa (%), and DECC ($10^{-2}$ km$^{-1}$) averaged over NCPR in
winter for the persistent years (red bars) and the reverse years (blue bars). (b)
Composite anomalies of the NAO index, wind speed at 850-hPa (m s$^{-1}$), relative
humidity at 850-hPa (%), and DECC (km$^{-1}$) averaged over NCPR in spring for the
persistent years (red bars) and the reverse years (blue bars).





The distinct evolutions of NCPR-average DECC, surface wind speed, relative
humidity, the winter EAWR index, and spring NAO index are summarized in Fig. 7.
In winter, positive phase of the EAWR teleconnection contributes to anticyclonic
anomalies over northeast Asia and associated southerly wind anomalies over the
NCPR, which further leads to positive DECC anomalies both in the persistent and
reverse years via reducing surface wind speed and increasing relative humidity (Fig.
7a). In spring, negative (positive) phase of the spring NAO contributes to formation of
the anomalous anticyclone (cyclone) over northeast Asia, and results in positive
(negative) DECC anomalies over the NCPR via increasing (decreasing) the relative
humidity and decreasing (increasing) the surface wind speed in the persistent (reverse)
years. Above evidences strongly indicate that different evolutions of atmospheric
anomalies over North Atlantic and mid-high latitude Eurasia plays a crucial role in the
distinct evolutions of the haze pollution over NCPR.

**4. Mechanism for the different evolutions of atmospheric anomalies over North**
**Atlantic and Eurasia**
What is the possible mechanism for the different evolutions of atmospheric
anomalies over North Atlantic and Eurasia? Considering that the internal atmospheric
process could not explain the connection of the atmospheric anomalies between two
seasons, the evolution of atmospheric anomalies over North Atlantic may be related to
the underlying SSTA. Previous studies have demonstrated that North Atlantic is the
region with strong air-sea interaction (Czaja et al. 2002; Czaja and Frankignoul 2002;



Huang and Shukla 2005; Pan 2005; Peng et al. 2003; Wu et al. 2009; Chen et al. 2016,
2018). On one hand, atmospheric anomalies over North Atlantic could lead to SSTA
via modulating surface heat fluxes (Czaja et al. 2002; Huang and Shukla 2005; Wu et
al. 2009; Chen et al. 2015). The connection between the atmospheric anomalies and
SSTA over North Atlantic is closest when atmospheric anomalies lead SSTA by about
one month (Czaja and Frankignoul 2002; Huang and Shukla 2005). On the other hand,
SSTA in the North Atlantic have a strong feedback on the overlying atmospheric
circulation via the heating-induced atmospheric Rossby wave response and the
interaction between low frequency mean flow and synoptic-scale eddy (Peng et al.
2003; Pan 2005; Czaja and Frankignoul 2002; Chen et al. 2020). In particular, a
number of studies have suggested that the development and evolution of atmospheric
anomalies and SSTA over North Atlantic are attributed to the positive air-sea
interaction process there (Czaja and Frankignoul 1999; Rodwell and Folland 2002;
Visbeck et al. 2003; Czaja et al. 2003; Wu and Liu 2005; Hu and Huang 2006; Chen
et al. 2019; Chen et al. 2020).

Evolutions of SSTA in the North Atlantic are examined in Fig. 8. In the persistent

years, significant cold SSTA are seen in the central North Atlantic around 30ºN and
off the east coast of Canada, together with notable warm SSTA in subtropical eastern
North Atlantic with a northeastward extension to the west coast of Europe (Fig. 8a).
The warm SSTA in the subtropical northeastern Pacific and the cold SSTA in the
central North Atlantic are maintained to the following spring with an increase in the
amplitude. In addition, high latitude North Atlantic is covered by warm SSTA in



spring. This forms a significant tripolar SSTA pattern in spring. Note that the tripolar
SSTA pattern is also the first EOF mode of interannual variation of SSTA in the North
Atlantic (not shown) (Chen et al. 2016, 2020). Studies have demonstrated that warm
(cold) SSTA in the tropical and subtropical North Atlantic related to the tripolar SST
anomaly pattern could induce a negative NAO-like pattern via the Rossby wave type
atmospheric response and wave-mean flow interaction process according to the
observational analysis and numerical experiments (Peng et al. 2003; Pan 2005; Czaja
and Frankignoul 2002; Chen et al. 2016, 2020).

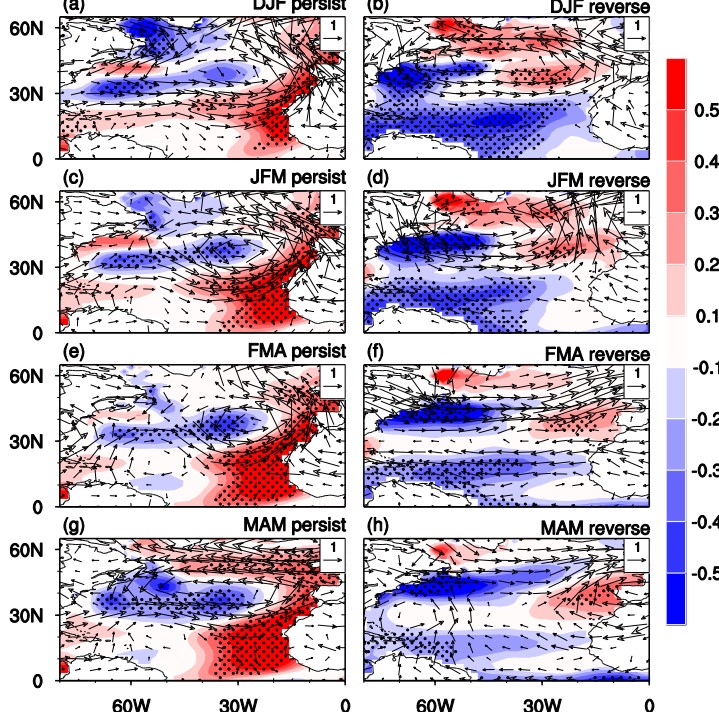


**Figure 8**. Composite anomalies of SST (°C) and 850-hPa winds (m s$^{-1}$) in (a, b)
D(0)JF(1), (c, d) JFM(1), (e, f) FMA(1), and (g, h) MAM(1) for (left column) the
persistent years and (right column) the reverse years. Stippling regions in the figure
indicate SST anomalies that are statistically significant at the 5% level.
In the reverse years, significant cold SSTA are seen in the tropical and
subtropical western North Atlantic in winter (Fig. 8b), which can maintain to
following spring with a decrease in the amplitude (Figs. 8h), which are in sharp
contrast to those in the persistent years (Figs. 8a, c, e, and g). It is reasonable to
speculate that the opposite SSTA in the tropical and subtropical North Atlantic may be
responsible for the opposite atmospheric anomalies over North Atlantic, which will be
confirmed later based on the linear barotropic model.

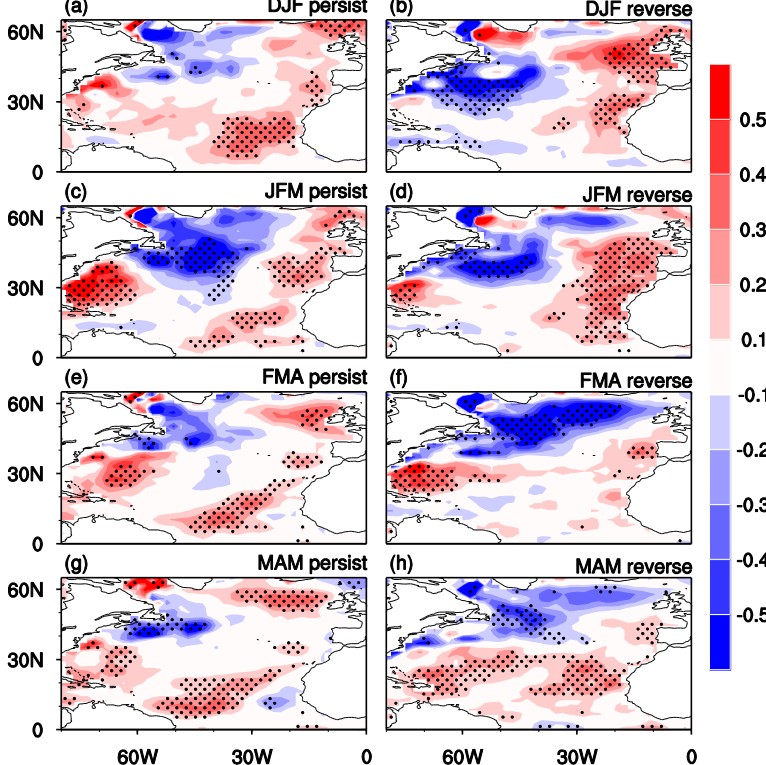


**Figure 9**. Composite anomalies of surface net heat fluxes (W m$^{-2}$) in (a, b) D(0)JF(1),
(c, d) JFM(1), (e, f) FMA(1), and (g, h) MAM(1) for (left column) the persistent years
and (right column) the reverse years. Stippling regions in the figure indicate
anomalies that are statistically significant at the 5% level.



530 Evolutions of SSTA in the North Atlantic from winter to the following spring are

531 related to the air-sea interaction. Figure 9 shows composite anomalies of the surface

532 net heat fluxes for the persistent and reverse years. Values of the surface heat fluxes

533 have been taken to be positive (negative) when their directions are downward

534 (upward), which contribute to warm (cold) SSTA. We have also examined composite

535 anomalies of SST tendency (not shown). It shows that spatial patterns of anomalies of

536 SST tendency in most parts of North Atlantic are similar to those of the surface net

537 heat fluxes anomalies. This suggests that changes in the surface net heat fluxes can

538 largely explain evolutions of SSTA in the North Atlantic from winter to the following

539 spring. For example, in the persistent years, significant positive net heat flux

540 anomalies are seen over the subtropical northeastern Atlantic from winter to spring

541 (Figs. 9a, 9c, 9e, and 9g), which could explain the formation and enhancement of the

542 positive SSTA there (Figs. 9a, 9c, 9e, and 9g). In addition, the negative surface net

543 heat flux anomalies to the east of the Canada explain generation and maintenance of

544 the negative SSTA there. Moreover, the positive surface net heat flux anomalies over

545 high latitudes contribute to warm SSTA. In the reverse years, positive net heat flux

546 anomalies appear off the west coast of west Europe (Figs. 9b, d, f, h), which explain

547 maintenance of the warm SSTA (Figs. 8b, d, f, h). In addition, positive net surface

548 heat flux anomalies over subtropical western North Atlantic in FMA and MAM (Figs.

549 9f and 9h) explain the decrease in the amplitude of the negative SSTA there (Figs. 8f

550 and 8h).

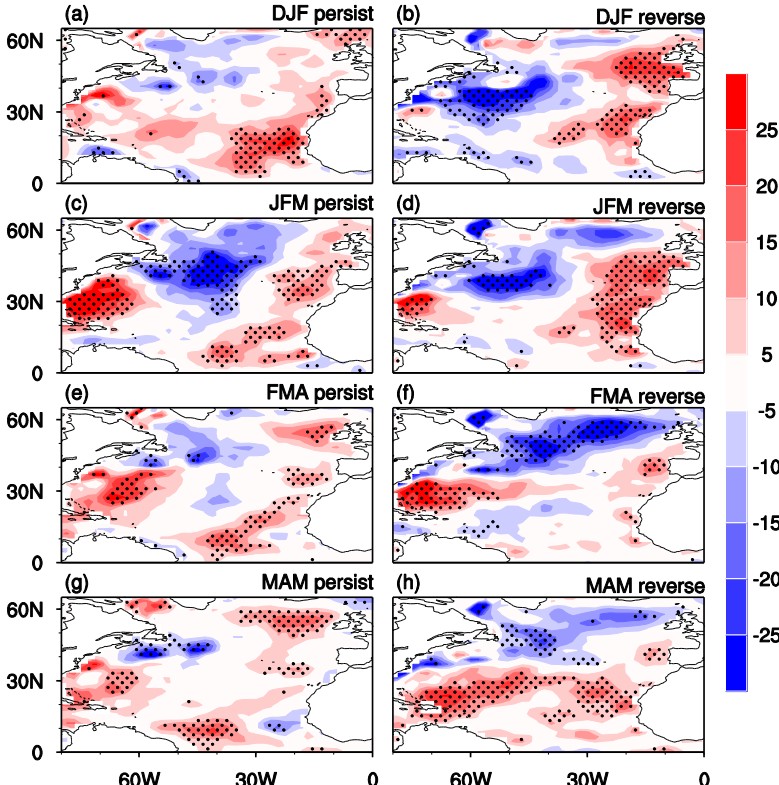

**Figure 10.** Composite anomalies of surface latent heat fluxes (W m$^{-2}$) in (a, b) D(0)JF(1), (c, d) JFM(1), (e, f) FMA(1), and (g, h) MAM(1) for (left column) the persistent years and (right column) the reverse years. Stippling regions in the figure indicate anomalies that are statistically significant at the 5% level.

Surface net heat flux anomalies are related to the overlying atmospheric circulation changes. Surface heat flux consists of four components, including the surface longwave and shortwave radiations, and surface latent and sensible heat fluxes. We find that surface net heat flux anomalies (Fig. 10) are dominated by changes in the surface latent heat flux (Fig. 10). Amplitudes of the surface sensible heat fluxes, and surface longwave and shortwave radiations are much weaker compared to that of the surface latent heat flux, and thus are not presented. In the persistent years, the



anomalous southwesterly winds over subtropical northeastern Atlantic in winter and
spring oppose the climatological northeasterly winds (Figs. 8a and 8g). This results in
decrease in the total wind speed and decrease in the upward latent heat flux (Figs. 10a
and 10g) and thus contribute to warm SSTA (Figs. 8a and 8g). Note that the warm
SSTA in the subtropical northeastern Atlantic could induce an anomalous cyclone to
its northwestward direction via Rossby wave type atmospheric response (Czaja and
Frankignoul 1999, 2002; Huang and Shukla 2005; Hu and Huang 2006; Chen et al.
2016, 2020) and help maintain the anomalous cyclone over mid-latitude North
Atlantic from winter to spring (Figs. 8a and 8g). Similarly, the anomalous easterly
winds along 60°N over North Atlantic oppose the climatological westerly winds (Figs.
8a and 8g), which lead to warm SSTA there via reduction of wind speed and upward
latent heat fluxes (Figs. 10a and 10g). By contrast, the anomalous northerly winds to
the western flank of the cyclonic anomaly bring colder and drier air from higher
latitude (Figs. 8a and 8g), which increase the upward latent heat flux and contribute to
cold SSTA (Figs. 10a and 10g). In the reverse years, southerly wind anomalies off the
west coast of west Europe carry warmer and wetter air northward from lower latitudes
and lead to warm SSTA (Figs. 8b and 8h) via reduction of upward latent heat flux
(Figs. 10b and 10h). In winter, northerly wind anomalies over the subtropical western
North Atlantic increase the trade wind (Fig. 8b), which result in enhancement of
surface latent heat flux (Fig. 10b) and partly contribute to cold SSTA (Fig. 8b). The
above analyses suggest that evolution of SSTA in the North Atlantic from winter to
subsequent spring is closely related to the air-sea interaction over the North Atlantic.





The notable differences in the SSTA in the tropical and subtropical North
Atlantic may explain the different atmospheric anomalies over North Atlantic and
Eurasia between the persistent and reverse years, with negative (positive) spring
NAO-like pattern and anticyclonic (cyclonic) anomaly over northeast Asia in the
persistent (reverse) years. Studies have demonstrated that springtime SSTA in the
tropical and subtropical North Atlantic have a strong impact on the atmospheric
circulation and associated climate anomalies over North Atlantic and Eurasia (Wu et
al. 2009; Wu et al. 2011; Chen et al. 2016, 2020). In particular, SSTA in the tropical
and subtropical regions could induce strong vertical motion and atmospheric heating
anomalies reaching to the upper-level troposphere (Ting 1996; Wu et al. 2009;
Hodson et al. 2010; Wu et al. 2011; Sun et al. 2015; Chen et al. 2020). Then, the
divergent/convergent anomalies at the upper-level troposphere induced by the SSTA
could be considered as effective sources for the generation of the atmospheric Rossby
wave (Watanabe 2004; Chen and Huang 2012; Zuo et al. 2013; Chen et al. 2020).
Considering that the atmospheric wave trains extending from the North Atlantic to the
Eurasia in Figs. 6c and 6d resemble an atmospheric stationary Rossby wave with an
equivalent barotropic vertical structure, the mechanism for their formation could be
examined based on the barotropic vorticity equation (Wu et al. 2011; Zuo et al. 2013;
Chen et al. 2016, 2020; O'Reilly et al. 2018). Hence, in the following, we perform
model simulations with barotropic model (Sardeshmukh and Hoskins 1988; Watanabe
2004; O'Reilly et al. 2018) to confirm the possible roles of the spring SSTA in the
North Atlantic in the formation of atmospheric anomalies over the North Atlantic and


Eurasia. Studies indicate that the barotropic model has a good performance in
capturing the key dynamics of the atmospheric response to the atmospheric heating
associated with the SSTA in the tropical and subtropical regions (Wu et al. 2011; Sun
et al. 2015; Zuo et al. 2013; Chen et al. 2016, 2020). Three experiments are performed:
the first experiment forced by the spring climatological mean vorticity (denoted as
EXP_Ctrl); the second experiment forced by the spring climatological mean vorticity
plus the given divergent anomalies over the subtropical northeastern Atlantic with a
center at 20°N, 20°W and maximum intensity of $7 \times 10^{-6} \times s^{-1}$ according to the spatial
pattern of spring SSTA in Fig. 8g (denoted as EXP_persist); the third experiment
forced by the spring climatological mean vorticity plus the given convergent
anomalies over the subtropical northwestern Atlantic with a center at 15°N, 60°W and
maximum intensity of $7 \times 10^{-6} \times s^{-1}$ according to the spatial pattern of spring cold
SSTA in tropical North Atlantic in Fig. 8h (denoted as EXP_reverse). Above three
experiments are integrated for 40 days. The barotropic model experiments can reach
equilibrium state quickly with only several days (Sardeshmukh and Hoskins 1988;
Zuo et al. 2013; Chen et al. 2016).

Figure 11a displays difference of atmospheric responses averaged during model

days 31-40 between EXP_persist and EXP_Ctrl with green contours representing the
prescribed divergent anomalies. In addition, difference of the responses between
EXP_reverse and EXP_Ctrl is exhibited in Fig. 11b. The barotropic model
experiments can well reproduce the distinct atmospheric anomalies between the
persistent and reverse years.

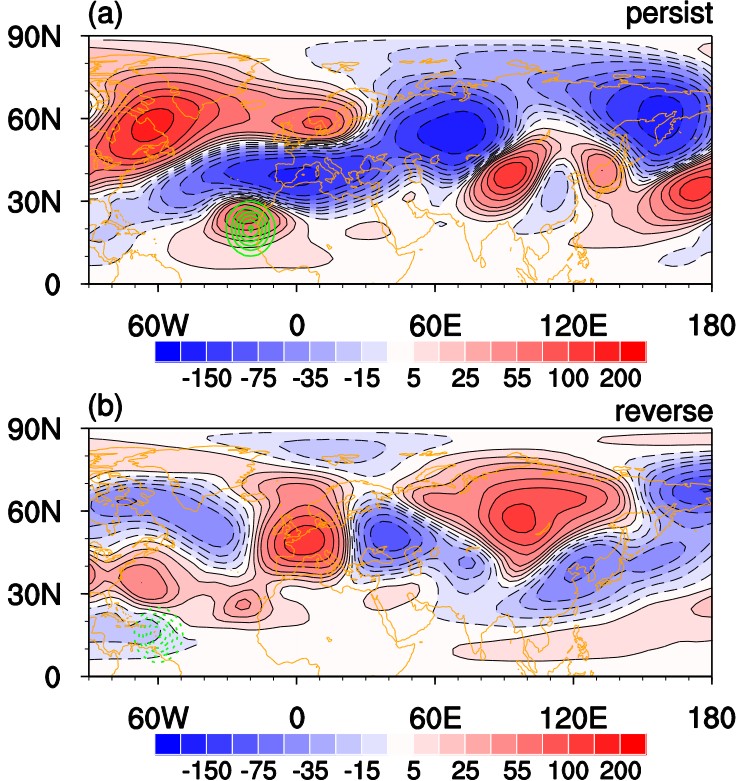


**Figure 11**. (a) Barotropic model height perturbation (unit: m) averaged from days 31
to 40 as a response to the given divergence anomaly (green contours with an interval
of $10^{-6}$ $s^{-1}$) over the subtropical eastern North Atlantic with the center at 20°N, 20°W.
(b) Barotropic model height perturbation (unit: m) averaged from days 31 to 40 as a
response to the given convergence anomaly (green contours with an interval of $10^{-6}$ $s^{-1}$)
over the subtropical western North Atlantic with the center at 15°N, 60°W.

In response to the prescribed divergent anomalies over the subtropical
northeastern Atlantic related to the warm SSTA there, there appears a positive
NAO-like pattern with negative geopotential anomalies over mid-latitudes (along
30°N) and positive anomalies over high-latitudes (along 60°N) North Atlantic
(Fig.11a), largely similar to the spatial pattern of spring atmospheric anomalies in the
persistent years in Fig. 6c. By contrast, in response to the prescribed convergent



anomalies over the subtropical northwestern Atlantic associated with the cold SSTA,
there exists a negative NAO-like pattern, with negative geopotential anomalies over
high-latitudes (along 60°N) and negative anomalies over mid-latitudes (along 30°N)
North Atlantic (Fig.11b), in concert with the spatial distribution of the atmospheric
anomalies in the reverse years in Fig. 6d. In addition, it is surprising to see that the
barotropic model experiment well simulate the anticyclonic (cyclone) anomaly over
northeast Asia and related southerly (northerly) wind anomalies over the NCPR in
response to the prescribed forcing in the subtropical northeastern (northwestern)
Atlantic as indicated in Fig. 11a (11b). This is consistent with the observed spring
atmospheric anomalies over East Asia for the persistent (reverse) years, although the
centers of the wave train over Eurasia in the barotropic experiments are not totally
identical to those in the observations. In general, the above barotropic experiments
further confirm the notion that the striking differences in the atmospheric anomalies
over North Atlantic and Eurasia (including northeast Asia) between the persistent and
reverse years can be attributable to the distinct SST anomalies in the North Atlantic.

## 5. Summary and discussions

This study examines different evolutions of haze pollution over NCPR from
winter to the succedent spring according to the analyses based on observational data
and reanalyses. It is found that interannual variation of haze pollution (as indicated by
the DECC) over NCPR in winter has a marginal positive relation with that in the
following spring, with a correlation coefficient of about 0.3 over 1980–2011 between



the haze pollution index in winter and spring, significant at the 90% confidence level.
This indicates that in most years when haze pollution over the NCPR is more (less)
serious in winter, air condition in the following spring is also worse (better) than
normal. Additionally, it is found that there appear some years when DECC anomalies
in the following spring are significantly opposite to those in winter. We then focus on
comparing atmospheric anomalies for the two types of years (i.e. persistent years and
reverse years) to understand why there occur two completely different evolutions of
haze pollution over the NCPR from winter to following spring, as schematically
summarized in Fig. 12.

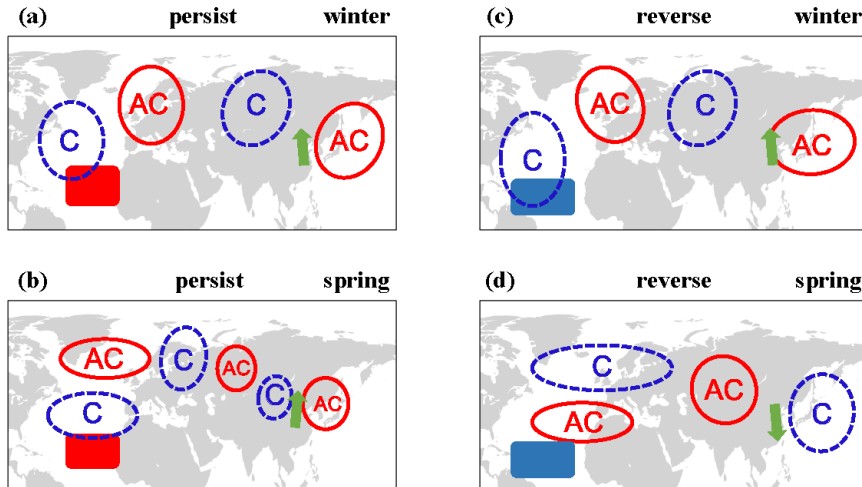


**Figure 12**. Schematic diagram showing evolutions of DECC, SST, and atmospheric
circulation anomalies from winter to spring for (left column) the persistent cases and
(right column) the reverse cases. Red solid contours (blue dashed contours) indicate
anticyclonic circulation anomalies (cyclonic circulation anomalies). Red (blue)
shadings in the North Atlantic indicate positive (negative) SST anomalies.

In the persistent years, above-normal DECC (indicating more serious haze

pollution) over the NCPR could be maintained to the succedent spring (Figs. 12a and



12b). This is attributable to the persistence of the anticyclonic anomaly over northeast
Asia and associated southerly wind anomalies to its west side over the NCPR (Figs.
12a and 12b). The southerly wind anomalies over the NCPR oppose the
climatological mean northerly winds, reduce the surface wind speed, and decrease the
horizontal dispersion of the pollutants, which finally lead to more serious haze
pollution in winter and spring. In addition, the southerly wind anomalies carry wetter
and warmer air from lower latitude, and lead to increase in the relative humidity,
which are also conducive to haze pollution. As have been demonstrated by previous
studies, the increase in the relative humidity is conducive to the generation of
secondary organic compounds and secondary aerosol species, which also has an
important contribution to the occurrence of haze pollution event over NCPR (Yu et al.
2005; Hennigan et al. 2008). Formation of the anticyclonic anomaly over the
northeast Asia in winter is closely related to the EAWR teleconnection pattern, while
in spring it is related to the positive phase of spring NAO and warm SSTA in the
subtropical northeastern Atlantic (Fig. 12a).
In the reverse years, an anticyclonic anomaly also appears over northeast Asia
and associated southerly wind anomalies occur over NCPR in winter, which
contribute to above-normal DECC (Fig. 12c). In addition, formation of the anomalous
anticyclone over the northeast Asia is also related to the EAWR pattern (Fig. 12c).
However, in the following spring, northeast Asia is covered by cyclonic anomaly
which is related to the positive phase of the NAO and cold SSTA in the subtropical
North Atlantic (Fig. 12d), which is in sharp contrast to those in the persistent years.



The northerly wind anomalies over the NCPR to the west flank of the anomalous
cyclone result in decrease in the DECC over the NCPR via reduction of relative
humidity and increasing the surface wind speed (Fig. 12d).
The distinct evolutions of atmospheric anomalies over North Atlantic and
Eurasia (including northeast Asia) are found to be closely related to the different
evolutions of SSTA in the North Atlantic. In the persistent (reverse) years, positive
(negative) SSTA in the subtropical northeastern (northwestern) Atlantic are
maintained to the following spring due to the positive air-sea interaction process.
Then, positive (negative) spring SSTA in the subtropical North Atlantic contribute to
the formation of negative (positive) NAO-like pattern over North Atlantic and the
generation of anticyclonic (cyclonic) anomaly over northeast Asia, and the occurrence
of associated southerly (northerly) wind anomalies over the NCPR via atmospheric
Rossby wave train. Results of barotropic model simulations with three experiments
further confirm the observed findings.
In this study, we find that negative SSTA in the subtropical northwestern Atlantic
play an important role for the formation of the positive NAO-like atmospheric
anomaly in the reverse years. It seems that wintertime surface heat flux changes
induced by the EAWR-related atmospheric anomalies cannot fully explain the
formation of strong cold SSTA in the subtropical northwestern Atlantic. This suggests
that other factors may also be important for the formation of the negative SST
anomalies, which remain to be explored. Studies indicated that ENSO-related SSTA in
the tropical Pacific also has a strong impact on atmospheric anomalies over East Asia





and haze pollution over eastern China (Wang et al. 2000; Li et al. 2017; Zhang et al.
2017; He et al. 2019). We have examined evolutions of SSTA in the tropical Pacific
from winter to subsequent spring in the persistent and reverse years. Results show that
SSTA in the tropical Pacific related to ENSO are weak both in the persistent and
reverse years (not shown). This suggests that ENSO-related SSTA may not have a
contribution to the interannual variation of haze pollution over the NCPR, which is
consistent with a recent study by He et al. (2019). It is reported that ENSO-related
SSTA in the tropical Pacific has a significant impact on the haze pollution over
southern China. By contrast, impact of ENSO on the haze pollution over North China
is weak (He et al. 2019). Furthermore, previous studies indicated that Arctic sea ice
and snow cover anomalies over Eurasia may also be important for the formation of
the atmospheric anomalies over East Asia in association with the haze pollution over
north China (Wang et al. 2015; Yin and Wang 2017). Whether snow cover and Arctic
sea changes play a role in contributing to the distinct evolutions of atmospheric
circulation anomalies over Eurasia and haze pollution over NCP remain to be
explored in the future.

**Code availability.** Figures in this study are constructed with the NCAR Command
Language (http://www.ncl.ucar.edu/). All codes used in this study are available from
the corresponding author (S.C.).

**Data availability**: Atmospheric data are derived from the NCEP-NCAR reanalysis



(http://www.esrl.noaa.gov/psd/data/gridded/data.ncep.reanalysis.html, last access: 6
February 2021) (NCEP-NCAR, 2021). SST data are obtained from the
https://psl.noaa.gov/data/gridded/data.noaa.ersst.v5.html (last access: 6 February 2021)
(NOAA, 2021). Atmospheric teleconnection indices are obtained from
https://www.cpc.ncep.noaa.gov/data/teledoc/telecontents.shtml (last access: 6
February 2021) (CPC, 2021). Surface data of visibility and relative humidity can be
obtained from the authors upon request.

**Author contributions.** Y.L. and C.S. designed the research, performed the analysis
and wrote the paper. All the authors discussed the results and commented on the
manuscript.

**Competing Interests.** The authors declare that they have no competing interests.

**Acknowledgments.** This work was supported jointly by the National Natural Science
Foundation of China (Grants 41721004 and 41961144025), and the Chinese Academy
of Sciences Key Research Program of Frontier Sciences (QYZDY-SSW-DQC024).

**Financial support**. This research has been supported by the National Natural Science
Foundation of China (Grants 41721004 and 41961144025), and the Chinese Academy
of Sciences Key Research Program of Frontier Sciences (QYZDY-SSW-DQC024).



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
