# Peer review of "Distinct evolutions of haze pollution from winter to following spring over the North China Plain: Role of the North Atlantic sea surface temperature anomalies"

_Atmospheric Chemistry and Physics, 2021_

## Author Response (AR1)

Reply to Reviewer#2 (comments in *italics*)

*Review of acp-2021-249*

*Haze pollution is a very serious environmental issue in China, especially in winter and spring. This manuscript discusses the possible connections of haze pollution in winter and spring. The authors show that conditions of the North Atlantic SST anomalies play a key role in determining seasonal evolutions of haze pollution over North China Plain. I find this study very interesting. The obtained results have important implications for the prediction of haze pollution. I suggest this manuscript to be accepted if satisfactorily addressing the following concerns.*

*General comments:*

*The authors define a DECC index by averaging the DECC anomalies over the 26 stations to describe variation of haze pollution over the NCPR. The authors need to prove that the DECC over NCPR varied similarly, i.e., the DEC of the 26 stations can be treated as a whole.*

**Reply:** Thanks for the Reviewer's suggestion. Following previous studies (Chen et al. 2019, 2020), the region for the NCPR we choose for analysis extends from 34ºN to 43ºN and from 114ºE to 120ºE. Chen et al. (2019) has demonstrated that interannual variations of the DECC at the 28 stations over the NCPR are largely similar, and can be treated as a whole. Figure R2A shows the first EOF mode (EOF1) of interannual anomalies of DECC in winter and spring over the NCPR (i.e. 34°-43°N and 114°-120°E) for the period of 1979-2010. Spatial structures of the EOF1 in winter and spring are generally featured by same-sign DECC anomalies over the NCPR, except for small patch of regions (Figs. R2Aa and R2Ab). In particular, the correlation coefficient between the PC time series corresponding to the EOF1 of winter DECC anomalies (Fig. R2C, red curve) and the winter NDI shown in Fig. 1b (in the revised manuscript) reaches 0.86, significant at the 99.9% confidence level. Similarly, the correlation coefficient between the PC time series corresponding to the EOF1 of spring DECC anomalies (Fig. R2C, blue curve) and the spring NDI shown in Fig. 1b (in the revised manuscript) is as high as 0.93. Above evidences suggest that the 28

stations in the NCPR can be generally considered as a whole. Figure R2A (below) and related discussions have been added in the revised manuscript. **Please see Figure 2 and Lines 241-254 in the revised manuscript.**

**References**

Chen, S., Guo, J., Song, L., Li, J., Liu, L., and Cohen, J.: Interannual variation of the spring haze pollution over the North China Plain: Roles of atmospheric circulation and sea surface temperature, Int. J. Climatol., 39, 783-798, 2019.

Chen, S., Guo, J., Song, L., Cohen, J., and Wang, Y.: Temporal disparity of the atmospheric systems contributing to interannual variation of wintertime haze pollution in the North China Plain, Int. J. Climatol., 40,128-144, 2020.

[Figure]

**Figure R2A.** The first EOF mode (EOF1) of interannual anomalies of DECC in (a) winter and (b) spring over the NCPR (i.e. 34°-43°N and 114°-120°E) for the period of 1979-2010. (c) The corresponding PC time series of EOF1 of interannual anomalies of DECC in winter (red curve) and spring (blue curve). The green dots in (a-b) indicate the stations in the NCPR.

*Lines 340-341 and Figure 5: I agree with the view that atmospheric circulation anomalies could exert impacts on haze pollution via modulating surface wind speed and humidity. Whether change in the boundary layer height (BLH) also plays a role? Studies have demonstrated BLH is also a very important factor in modulating Haze pollution via change in the vertical diffusion of pollutant. For example, the anticyclonic anomaly and associated increase in sea level pressure over the North China plain may lead to decrease in the BLH, which would result in more serious haze pollution. The role of the BLH should be examined.*

**Reply**: Thanks for the Reviewer's suggestion. Following your suggestion, we have examined anomalies of the boundary layer height (BLH). Figure R2B (below) shows composite anomalies of BLH in winter and spring for the persistent and reverse years. Consistent with the Reviewer's view, it shows that decrease in the BLH, which is associated with the anticyclonic anomaly, is seen over the NCPR in winter and spring for the persistent year (Figs. R2Ba and R2Bc). Decrease in the BLH is unfavorable for vertical diffusion of pollutant and thus contribute to maintenance of the above normal DECC from winter to spring for the persistent year (Figs. 3a and 3c in the revised manuscript). By contrast, in the reverse year, the winter decreased BLH (Fig. R2Bb) is replaced by increased BLH in the subsequent spring over the NCPR (Fig. R2Bd). Increase in the BLH in spring over the NCPR contributes to less serious haze pollution via increase in the vertical diffusion of pollutant. Following the Reviewer's suggestion, we have discussed the role of the BLH anomalies in the different evolutions of haze pollution from winter to spring. **Figure R2B (below) have been added to the revised manuscript. Please see Figure 7 in the revised manuscript.**

[Figure]

**Figure R2B.** Composite anomalies of boundary layer height (BLH, unit: m) in (a, b) winter and (c, d) spring in the (left column) persistent years and the (right column) reverse years. Stippling regions indicate anomalies that are statistically significant at the 5% level.

*This study reported that North Atlantic SST anomalies play a key role in the formation of the atmospheric circulation anomalies via atmospheric teleconnection, which further determine evolutions of haze pollution over North China. From Fig. 6, it seems that geopotential height anomalies in the Arctic region also show large differences. Previous studies have shown that Arctic sea ice anomalies can significantly impact atmospheric circulation anomalies over East Asia and haze pollution over China. Hence, I suggest authors examine Arctic sea ice anomalies and discuss whether Arctic sea ice conditions also play a role in the different evolution of atmospheric circulation anomalies and haze pollution.*

**Reply**: Thanks for the Reviewer's suggestion. Following your suggestion, we have examined evolutions of Arctic sea ice anomalies for the persistent and reverse years

(Fig. R2C below). From Fig. R2C, sea ice anomalies in winter and spring for the persistent and reverse years are weak over most portions of the Arctic. This suggests that the distinct evolutions of atmospheric circulation and haze pollution over the NCPR for the persistent and reverse years are not likely due to the Arctic sea ice anomalies. We have added discussions in the revised manuscript. **Please see Lines 777-782 in the revised manuscript.**

[Figure]

**Figure R2C.** Composite anomalies of Arctic sea ice concentration (unit: %) in (a, b) winter and (c, d) spring in the (left column) persistent years and the (right column) reverse years. Stippling regions indicate anomalies that are statistically significant at the 5% level.

*I suggest add associated wave activity flux into Figure 11 to more clearly illustrate propagation of the atmospheric wave train induced by the forcing over the North*

*Atlantic.*

**Reply**: Following the Reviewer's suggestion, we have added the associated wave activity fluxes to Figure 13 (in the revised manuscript, or please see Figure R2D below) to more clearly illustrate propagation of the atmospheric wave train.

[Figure]

**Figure R2D.** Barotropic model height perturbation (unit: m) averaged from days 31 to 40 as a response to the given divergence anomaly (green contours with an interval of $10^{-6}$ s$^{-1}$) over the subtropical eastern North Atlantic with the center at 20°N, 20°W. (b) Barotropic model height perturbation (unit: m) averaged from days 31 to 40 as a response to the given convergence anomaly (green contours with an interval of $10^{-6}$ s$^{-1}$) over the subtropical western North Atlantic with the center at 15°N, 60°W. Vectors in (a)-(b) indicate the corresponding wave activity fluxes.

*The barotropic experiment simulations can well confirm the observed results. It is interesting. My question is: why the barotropic experiment simulation is only integrated for 40 days? In addition, why selected 31-40 days to analyze? Why not*

*selected 25-35 days or other days to analyze?*

**Reply**: Thanks for the Reviewer's suggestion. As indicated by previous studies (Sardeshmukh and Hoskins, 1988; Watanabe, 2004; Wu et al., 2010; Chen et al. 2016, 2019), experiments of the barotropic model only need several days to reach balance. Therefore, following many previous studies (e.g., Wu et al., 2010; Chen et al., 2016, 2019, etc), the average of 31-40 days were selected (as the barotropic model should have reach balance for model day 31-40). Figure R2E display the results for the average of 25-35 days. It shows that the atmospheric anomalies for the average of 31-40 days (Figure 13 in the revised manuscript) are similar to those average of 25-35 days (Figure R2E below). We have added discussions in the revised manuscript.

**Please see Lines 661-666 in the revised manuscript.**

**References**

Chen, S., Wu, R., and Liu, Y.: Dominant modes of interannual variability in Eurasian surface air temperature during boreal spring, J. Clim., 29, 1109–1125, 2016.

Chen, S., Guo, J., Song, L., Li, J., Liu, L., and Cohen, J.: Interannual variation of the spring haze pollution over the North China Plain: Roles of atmospheric circulation and sea surface temperature, Int. J. Climatol., 39, 783-798, 2019.

Watanabe, M.: Asian jet waveguide and a downstream extension of the North Atlantic Oscillation, J. Clim., 17, 4674–4691, 2004.

Wu, R., Yang, S., Liu, S., Sun, L., Lian, Y., and Gao, Z.: Northeast China summer temperature and North Atlantic SST, J. Geophys. Res., 116, D16116, 2011.

Sardeshmukh, P. D., and Hoskins, B. J.: The generation of global rotational flow by steady idealized tropical divergence, J. Atmos. Sci., 45, 1228–1251, 1988.

[Figure]

**Figure R2E**. As in Fig. R2D, but for the averaged from days 25 to 35.

*Specific comments:*

*Line 53-54: the occurrences of haze pollution event -> the occurrence of haze pollution events*

**Reply**: Modified as suggested.

*Please re-plot Fig 2(c), as there is a text spelling mistake (presist->persist).*

**Reply**: Thanks for pointing this out. We have modified the related Figure.

*Line 308: winds anomalies -> wind anomalies*

**Reply**: Modified.

*Line 358: leads to -> and leads to*

**Reply**: Modified as suggested.

*Line 410: also resembles -> and also resembles*

**Reply**: We have modified it.

*Line 413: similar region-> same region?*

**Reply**: Modified as suggested.

*Line 421: have -> has*

**Reply**: Modified.

*Line 466: leads -> lead*

**Reply**: Modified.

*Line 473: plays -> play*

**Reply**: Modified.

*Line 488: closest -> the closest*

**Reply**: Modified as suggested.

*Please check the references carefully, such as Wang et al. 2014 in line 64 is not found in the references. In addition, it is better to arrange the references in alphabetical order.*

**Reply**: Thanks for pointing this out. We have checked the references carefully. In addition, we have arranged the references in the alphabetical order following the Reviewer's suggestion.

Reply to Reviewer#3 (comments in *italics*)

*The authors linked the evolution of winter-spring haze pollution in the North China Plain to the SST anomalies over the North Atlantic sector. They demonstrated this linkage through composition analysis and a simple barotropic model. Seasonal prediction of air pollution is of great importance for the sake of public health. I think this manuscript's scope fits the ACP journal.*

**Reply:** We thank the Reviewer very much for your comment and considering our manuscript fits the scope of ACP.

*While the conclusion of this manuscript seems reasonable, I am not convinced about the scientific novelty of this study. Moreover, the authors have reported the role of North Atlantic SST in spring haze by Chen et al. (2019).*

*Chen, S., Guo, J., Song, L., Li, J., Liu, L., and Cohen, J.: Interannual variation of the spring haze pollution over the North China Plain: Roles of atmospheric circulation and sea surface temperature, Int. J. Climatol., 39, 783-798, 2019.*

**Reply:** Thanks for the Reviewer's comment.

We acknowledge that our previous study (Chen et al. 2019, this reference had been mentioned in detail in the introduction) has reported the important role of the North Atlantic SST anomalies in modulating spring haze pollution over the North China Plain. However, **the purpose of this study is totally different from that of Chen et al. (2019)**. Chen et al. (2019) examined the factors for the interannual variation of haze pollution over the North China Plain in spring. In sharp contrast, this study examined the evolution of haze pollution from preceding winter to following spring. To the best of our knowledge, none of previous study has examined the across-season evolutions of haze pollution over the North China Plain from winter to spring. Impacts of haze pollution depend strongly on its evolution (the long-lasting haze pollution should have more severe impacts).

In addition to different motivations, **the present study obtained some new findings, which is a further development of Chen et al. (2019)**. We found that interannual variation of haze pollution over North China Plain in winter has a positive

relation with that in the following spring, with a correlation coefficient of 0.3 between the haze pollution index in winter and spring for 32 years, which is significant at the 90% confidence level according to the two-tailed Student's $t$ test. That is to say, in most years when haze pollution over the North China Plain is more (less) serious in winter, air condition in the following spring is also worse (better) than normal. To our knowledge, this new finding has not been reported in any of previous studies. We then examine the factors contributing to the across-season persistence of haze pollution over the North China Plain from winter to succedent spring, and confirm that North Pacific SST plays an important role (the factor is same as Chen et al. 2019). Furthermore, as the correlation coefficient is not so high (can only pass the 90% confidence level according to the two-tailed Student's $t$ test), we found there actually exist two very different seasonal evolution types of haze pollution over the North China Plain. These two types of years (i.e. persistent years and reverse years) were then examined in detail to understand why there occur two completely different evolutions of haze pollution over the North China Plain from winter to following spring. Results of the present study demonstrated that the distinct evolutions of atmospheric anomalies over North Atlantic and Eurasia are closely related to the different evolutions of SSTA in the North Atlantic. Moreover, we have examined the physical processes responsible for distinct evolutions of SST anomalies in the North Atlantic from winter to spring in detail via an analysis of surface heat budget.

Last but not least, **the meaning and value of the present research is different from and progressive of Chen et al. (2019)**. We would like to emphasize that Chen et al. (2019) mainly examined the simultaneous relationship between spring North Atlantic SST anomalies and spring haze pollution over the North China Plain (please see their Fig. 10), and thus the results of Chen et al. (2019) actually give a reasonable explanation of physical mechanism but have no implication for the prediction of the spring haze pollution over the North China Plain. For example, if the haze pollution over the North China Plain is serious in this coming winter (i.e. Winter 2021-22), we do not know whether air condition over the North China Plain in the following spring (hereinafter Spring 2022) will get better or worse only solely based on the results of

Chen et al. (2019). By contrast, results obtained in this study have important implications for the seasonal prediction of haze pollution over the North China Plain. For instance, similarly, if we know that the haze pollution over the North China Plain is serious in this coming winter, we can predict the air condition in the following spring by one-season ahead based on this study. In particular, if the SST anomalies in the tropical North Atlantic are above (below) normal in Winter 2021-22, we could then could predict that haze pollution in the following Spring 2022 is most probably to be more (less) serious according to the findings of this study. Improving the seasonal prediction of haze pollution is of critical importance due to substantial economic losses, severe harm to human health and many premature deaths caused by haze pollution.

In summary, as the reviewer suggested, the important role of the North Atlantic SST anomalies in the variation of spring haze pollution as well as the physical process for the impact of the North Atlantic SST anomalies are similar between this study and Chen et al. (2019). However, their research motivations, main findings and potential implications are different between each other. Based on the conclusions of Chen et al. (2019) that revealed the factors for the variation of haze pollution in simultaneous spring, this manuscript is the first to explore the seasonal evolutions of haze pollution from winter to following spring over the North China Plain, and is also the first to find two different evolution types (persistent and reverse) as well as the associated physical processes..

**Please see Lines 111-124 in the revised manuscript for the motivation of this study.**

*Specifically, it is a good idea to focus on the seasonal evolution of haze pollution. But I didn't find the relationship between winter haze pollution and spring haze pollution, and Figure 7 shows that leading meteorological factors driving haze pollution are identical in both persist and reverse winters. As such, I think the finding in this study is only about what drivers spring haze. Consequently, the main conclusions in this study are very similar with the previous one (Chen et al., 2019).*

**Reply:** Thanks for the Reviewer's comment.

We agree with the Reviewer's view that the role of the North Atlantic SST anomalies in the variation of haze pollution over the North China Plain in spring is similar between this study and Chen et al. (2019). Results of this study can confirm the findings of Chen et al. (2019).

Although the key driven factor is found and verified to be the same (i.e., the North Atlantic SST anomalies), we can gain more new understanding from the present study. What leads to serious Haze pollution over North China Plain in spring? Chen et al. (2019) answered this question, indicating that the North Atlantic SST in spring plays an important role. Then, we broaden our horizons to winter instead of spring. Does the North Atlantic SST in winter play a role for Haze pollution in winter? The answer is No, which is different from Chen et al. (2019). Then, we further broaden our horizons to both winter and the following spring. Whether there exists a relation between interannual variation of haze pollution over the North China Plain in winter and following spring? What contributes to the seasonal persistence of Haze pollution over the North China Plain from winter to spring? Obviously, Chen et al. (2019) is not the results of these questions, but can only be served as one of the footstones to answer these questions.

In addition, the purposes, the findings and the implication for prediction of this manuscript are all different from those of Chen et al. (2019), please see the detailed Reply above.

*Statistically, the correlation coefficient between the winter and spring haze over 1980–2011 is only 0.30 (at the 90% confidence level), with a total of 11 (9) years for the in-phase (out-of-phase). This indicates that winter and spring haze pollution are not well connected.*

**Reply:** We agree with the Reviewer's view that the correlation coefficient between the winter and spring haze indexes over 1980-2011 is not very high, which is only 0.30. However, it can pass the 90% confidence level according to the two-tailed Student's $t$ test, which indicates a significant relationship from a statistical point of

view but still remains considerable uncertainty as the reviewer pointed out. Hence, the occurrence of in-phase and out-of-phase between the winter and spring haze over 1980-2011 is further examined. It is found that 20 out of 32 years with positive/negative winter Haze pollution anomalies are followed by positive/negative Haze pollution anomalies in the following spring over the North China Plain (Fig. 1b). This indicates that the portion is about 62.5% for the persistent relationship between the winter and spring haze. Moreover, the correlation coefficient between the winter and spring haze is as high as 0.71 after exclusion of the extreme reverse years (a total of 9 reverse years, Please see Table 1 in the revised manuscript), while it is only -0.10 after exclusion of the extreme persistent years (a total of 11 reverse years, Please see Table 1 in the revised manuscript). Hence, it suggests a significant positive correlation relationship between the winter and spring haze, and also implies an interesting out-of-phase phenomenon that is crucial for cross-season prediction of spring haze. In addition, Monte Carlo method is adopted to evaluate the robustness of the correlation coefficient by constructing 10,000 random realizations of spring haze time series (Fig. R3A below). The correlation coefficient with permutation of 90% (95%) confidence is about 0.23 (0.3) (Fig. R3A). Note that very similar results can be obtained if we construct 100,000 random realizations (Fig. R3B below). Therefore, the Monte Carlo permutation tests demonstrate that the linkage between the winter and spring haze can pass the 95% confidence level. Hence, their linkage is not random, but is significant and meaningful.

**We have added several related descriptions in the revised manuscript. Please see Lines 255-271 in the revised manuscript.**

[Figure]

**Figure R3A.** Significance level of the correlation coefficient between winter Haze index and spring Haze index estimated by the Monte Carlo method. Particularly, the Monte Carlo method is adopted to evaluate the robustness of the correlation coefficient by constructing (a) 10,000 random realizations, and (b) 100,000 random realizations of spring haze time series. Y-axis indicates the correlation coefficient between the winter Haze index and the randomly generated realizations of the spring haze index. Horizontal red lines indicate the correlation coefficients of 0.23 and 0.3, significant at the 90% and 95% confidence level based on the Monte Carlo method.

*I am sorry that I can't be more positive at this time. I encourage the authors to do further analysis if they believe this seasonal linkage.*

**Reply:** Please see the replies above. We thanks very much again for the Reviewer's valuable and encouraging comment.